# Synergistic Combination of AS101 and Azidothymidine against Clinical Isolates of Carbapenem-Resistant *Klebsiella pneumoniae*

**DOI:** 10.3390/pathogens10121552

**Published:** 2021-11-29

**Authors:** Chung-Lin Sung, Wei-Chun Hung, Po-Liang Lu, Lin Lin, Liang-Chun Wang, Tsung-Ying Yang, Sung-Pin Tseng

**Affiliations:** 1Department of Medical Laboratory Science and Biotechnology, College of Health Sciences, Kaohsiung Medical University, Kaohsiung 807, Taiwan; jimmy0922236561@gmail.com; 2Department of Microbiology and Immunology, College of Medicine, Kaohsiung Medical University, Kaohsiung 807, Taiwan; wchung@kmu.edu.tw; 3Center for Liquid Biopsy and Cohort Research, Kaohsiung Medical University, Kaohsiung 807, Taiwan; d830166@gmail.com; 4Division of Infectious Diseases, Department of Internal Medicine, Kaohsiung Medical University Hospital, Kaohsiung 807, Taiwan; 5School of Post-Baccalaureate Medicine, College of Medicine, Kaohsiung Medical University, Kaohsiung 807, Taiwan; 6Department of Culinary Art, I-Shou University, Kaohsiung 84001, Taiwan; lynnlin@isu.edu.tw; 7Department of Marine Biotechnology and Resources, National Sun Yat-sen University, Kaohsiung 804, Taiwan; marknjoy@g-mail.nsysu.edu.tw; 8Drug Development and Value Creation Research Center, Kaohsiung Medical University, Kaohsiung 807, Taiwan; 9Graduate Institute of Animal Vaccine Technology, College of Veterinary Medicine, National Pingtung University of Science and Technology, Pingtung 912, Taiwan

**Keywords:** carbapenem-resistant Enterobacteriaceae (CRE), synergistic activity, AS101, azidothymidine

## Abstract

Owing to the over usage of carbapenems, carbapenem resistance has become a vital threat worldwide, and, thus, the World Health Organization announced the carbapenem-resistant Enterobacteriaceae (CRE) as the critical priority for antibiotic development in 2017. In the current situation, combination therapy would be one solution against CRE. Azidothymidine (AZT), a thymidine analog, has demonstrated its synergistically antibacterial activities with other antibiotics. The unexpected antimicrobial activity of the immunomodulator ammonium trichloro(dioxoethylene-*o*,*o*’)tellurate (AS101) has been reported against carbapenem-resistant *Klebsiella pneumoniae* (CRKP). Here, we sought to investigate the synergistic activity between AS101 and AZT against 12 CRKP clinical isolates. According to the gene detection results, the *bla*_OXA-1_ (7/12, 58.3%)_,_ *bla*_DHA_ (7/12, 58.3%), and *bla*_KPC_ (7/12, 58.3%) genes were the most prevalent ESBL, AmpC, and carbapenemase genes, respectively. The checkerboard analysis demonstrated the remarkable synergism between AS101 and AZT, with the observable decrease in the MIC value for two agents and the fractional inhibitory concentration (FIC) index ≤0.5 in all strains. Hence, the combination of AS101 and azidothymidine could be a potential treatment option against CRKP for drug development.

## 1. Introduction

Carbapenem-resistant Enterobacteriaceae (CRE) was announced as the critical priority for antibiotic development in 2017 [1]. Recently, the COVID-19 pandemic promoted the spread of carbapenem-resistant *Klebsiella pneumoniae* (CRKP) [2], highlighting an urgent need for novel treatment options. The high carbapenem-resistant rates were revealed in some parts of Europe [3], especially Southern Europe. According to the European Centre for Disease Prevention and Control EARS-Net, 58.3% of 312 Greek *K. pneumoniae* isolates collected in 2019 were resistant to either imipenem or meropenem [4]. In Romania, 32.3% of 470 *K. pneumoniae* were reported as carbapenem-resistant isolates; in Italy, 28.5% of 7327 *K. pneumoniae* were defined as carbapenem-resistant isolates. A regional resistance surveillance program in Asia-Pacific reported 25% carbapenem-resistant rate in *K. pneumoniae* for India and 5% for both the Philippines and Thailand [5]. A more recent surveillance conducted in Taiwan, 2018, described a 7.2% carbapenem-resistant rate in 346 *K. pneumoniae* isolates [6]. In view of the aforementioned reports, the development of novel therapeutic options to address CRKP is urgently needed.

In recent years, some novel β-lactam-β-lactamase inhibitor combination therapies have been launched in clinical settings, for instance, meropenem–vaborbactam, imipenem–relebactam, and ceftazidime–avibactam [7,8,9,10]. Moreover, azidothymidine was reported to show synergistic activities with some clinical antimicrobial agents such as colistin and fosfomycin. [11,12,13]. Azidothymidine (AZT), a thymidine analog, possesses an antiretroviral activity and clinically treats patients with human immunodeficiency virus (HIV) infections, the acquired immunodeficiency syndrome (AIDS) [14]. Lately, studies have reported that azidothymidine combined with antibiotics produced synergistic activities against antibiotic-resistant gram-negative bacteria both in vitro and in vivo [11]. A previous study also reported that a combination of colistin and azidothymidine revealed synergistic activity against colistin-resistant CRKP clinical isolates [12]. In a phase I clinical trial for the combination of AZT and colistin, the results revealed that a dosage of 2 million IU CMS plus 100 mg AZT twice a day might be sufficient for urinary tract infections (UTIs) [15]. Although the phase I clinical trial of the combination of AZT and colistin is in progress, colistin could be a temporary option for CRKP treatment but could not be permanent because of its nephrotoxicity and neurotoxicity [16].

AS101 is a fully-synthesized, tellurium-containing, organic compound with a small molecular weight of 312 Daltons [17,18]. With its characteristic of immunomodulation, AS101 was used to treat autoimmune diseases, inflammatory bowel disease (IBD), multiple sclerosis (MS), and psoriasis [19], and some of its applications are in ongoing clinical trials. Moreover, AS101 demonstrated anti-inflammatory activity [19], antiviral activities (HIV-1 and WNV) [20,21], and antimicrobial activities (carbapenem-resistant *Acinetobacter baumannii*, *K. pneumoniae*, *Enterobacter cloacae*) [22,23,24,25]. In a recent study, Yang et al. demonstrated its in vitro and in vivo antimicrobial activity against carbapenem-resistant *Acinetobacter baumannii* [22]. Although previous efforts posed that AS101 could be a potential option to treat CRKP, our study described that the MIC values of AS101 against CRKP was up to 32 μg/mL [23], which is not far away from its 50% cytotoxicity level (145 μg/mL) [21]. Hence, it might be unsafe for high dose. To this end, some improvements might be needed to increase the antibacterial activity of AS101. Accordingly, finding an alternative agent to combine with AZT for CRKP treatment is of vital importance. Given the previous efforts on AZT combinations and AS101, we sought to evaluate the synergistic activities of AZT plus AS101 against CRKP in this study.

## 2. Results

### 2.1. Antimicrobial Susceptibility Testing

Among the 12 isolates tested in this study, over 80% of resistant rates were observed in 16 agents (Table 1): ampicillin (12/12, 100%), aztreonam (10/12, 83.3%), ceftazidime (11/12, 91.7%), cefazolin (12/12, 100%), ciprofloxacin (12/12,100%), cefepime (10/12, 83.3%), cefoxitin (12/12 100%), ceftriaxone (12/12, 100%), cefotaxime (11/12, 91.7%), imipenem (12/12, 100%), levofloxacin (12/12, 100%), meropenem (12/12, 100%), trimethoprim/sulfamethoxazole (11/12, 91.7%), piperacillin–tazobactam (11/12, 91.7%), ertapenem (12/12, 100%), doripenem (12/12, 100%). In contrast to the 16 agents with low susceptibilities, tigecycline, amikacin, and gentamicin had higher susceptibilities of 91.7% (11/12), 66.7% (8/12), and 41.7% (5/12), respectively.

### 2.2. Detection of Resistance Gene

Among the 12 CRKP isolates, the *Klebsiella pneumoniae* carbapenemase (KPC) gene was the dominant carbapenemase gene detected in seven isolates (7/12, 58.3%), followed by *bla*_OXA-48_ gene in one isolate (1/12, 8.3%) (Figure 1). Four of the isolates (4/12, 33.3%) were found with only one type of β-lactamase genes, mostly *bla*_KPC_ gene (three isolates). Nevertheless, only one of the isolates (1/12, 8.3%) had two types of resistant gene (*bla*_OXA-48_ and *bla*_OXA-1_). Two of the isolates (2/12, 16.7%) harbored three types of resistant gene, including either *bla*_OXA-1_ or *bla*_CTX-M_ in one of the isolates, and *bla*_DHA_ and *bla*_TEM_ in both. In addition, five isolates (5/12, 41.6%) were found with four types of β-lactamase genes simultaneously. All of these five isolates harbored *bla*_OXA-1_ and *bla*_DHA_. Moreover, four of these five isolates harbored *bla*_SHV-12_ and *bla*_KPC_. The *bla*_DHA_ gene was detected in seven isolates, but none of the isolates harbored *bla*_CMY_. The *bla*_OXA-1_ gene was the most prevalent of the ESBL genes (7/12, 58.3%) detected in this study, followed by *bla*_TEM_ (4/12, 33.3%), *bla*_SHV-12_ (4/12, 33.3%), and *bla*_CTX-M_ (2/12, 16.7%).

### 2.3. Evaluation of Synergistic Effects

The MIC values for AS101 against the 12 CRKP isolates ranged from 2 to 512 μg/mL (Table 2), with the MIC_50_, MIC_75_, and MIC_90_ of 128, 256, and 512 μg/mL, respectively. The MIC range of azidothymidine (AZT) against the 12 CRKP was from 0.5 to 4 μg/mL, and the MIC_50_, MIC_75_, and MIC_90_ were revealed as 1, 2, and 2 μg/mL, respectively. With the combination of AS101 and AZT, noticeable decreases for MIC_50_, MIC_75_, and MIC_90_ were observed in both AS101 (from 128, 256, and 512, respectively, to 8, 16, and 16 μg/mL) and AZT (from 1, 2, and 2, respectively, to 0.25, 0.25, and 0.5 μg/mL) (Table 2). The MIC distributions of AS101 and AZT alone or in combination are visualized in Figure 2. Significant changes in MIC distributions were noted for AS101 and AZT (both *p* < 0.0001), with decreased MIC ranges for AS101 (from 2–12 to 0.5–32 μg/mL) and AZT (from 0.5–4 to 0.0625–1 μg/mL) (Table 2). The fractional inhibitory concentration (FIC) indexes of the 12 isolates were all ≤0.5 (Table 3), suggesting the synergistic interaction between AS101 and azidothymidine. Additionally, the synergistic effect was observed in the growth curve (Appendix A.) Supported by the aforementioned results, the combination therapy of AS101 and AZT is a potential treatment option for CRKP infections, needing further investigation in future study. All of the checkerboard methods for checking the synergistic effect were performed and repeated three times, and all results were reproducible.

## 3. Discussion

Due to the transmission of resistant genes, the prevalence of carbapenem-resistant Enterobacteriaceae (CRE) has become a deadly threat to public health in the past years, causing severe infections associated with significant mortality [26]. Among approximately 140,000 cases of healthcare-associated Enterobacteriaceae infections in the United States yearly, around 9300 instances were caused by CRE, and 520 patients infected by CRE died [27]. The multidrug resistance was usually noticed among CRE isolates [28,29,30,31]. In a previous study conducted in the USA, a larger amount of CRKP isolates were collected from Jan 2014 to Mar 2015 in a long-term acute care hospital network [30]. The susceptibilities of selected antibiotics were examined against these isolates, including amikacin, ciprofloxacin, levofloxacin, gentamicin (or tobramycin), colistin (or polymyxin B), and tigecycline. The susceptibility to tigecycline was found to be the highest (413/439, 94.1%), followed by colistin (or polymyxin B) (579/690, 83.9%), and amikacin (298/885, 33.7%). Low susceptibilities were revealed for ciprofloxacin (10/630, 1.6%), levofloxacin (12/713, 1.7%), and gentamicin (or tobramycin) (11/630, 1.7%). In Taiwan, Chiu et al. collected 457 isolates of CRKP from 21 hospital between Jan 2012 and Aug 2015 [28], and antimicrobial susceptibility testing of 19 agents against the 457 isolates was performed. Among agents they tested, only amikacin, colistin, and tigecycline showed high potencies, with susceptibilities of 78.8% (360/457), 85.6% (391/457), and 88.6% (405/457), respectively. According to a recent study from Iran, 50 isolates of CRKP were examined against 15 agents and were revealed with an 85% susceptibility rate against only amikacin and a high resistance rate against the other antimicrobial agents [31]. In this study, tigecycline demonstrated the highest susceptibility (91.7%, 11/12) among 19 agents, followed by amikacin (66.7%, 8/12) and gentamicin (41.7%, 5/12), highlighting an urgent need for novel agents for CRKP.

Owing to the possibility of the horizontal transfer, the plasmid-mediated carbapenem resistance, namely the carbapenemase (gene), has been regarded as a critical mechanism needed to be monitored [32,33]. A previous study on carbapenemase-producing CRE isolates between 2013 and 2016 in the USA illustrated a diversity of carbapenemase genes, with *bla*_OXA-48_ (25.0%, 6/24), *bla*_KPC_ (20.8%, 5/24), *bla*_NDM_ (20.8%, 5/24), *bla*_SME_ (20.8%, 5/24), *bla*_IMP_ (8.3%, 2/24), and *bla*_VIM_ (4.2%, 1/24) [33]. A molecular epidemiological study from Europe revealed that a *bla*_KPC_-like gene was the dominant carbapenemase gene in Italy, Greece, Portugal, Israel, and the UK, with a *bla*_NDM_-like gene in Serbia. However, the dominant carbapenemase gene in Turkey, Spain, Romania, and Belgium was a *bla*_OXA-48_-like gene [34]. In a report from the Middle East, Alizadeh et al. described that *bla*_OXA-48_ was the most prevalent carbapenemase gene among 50 CRKP isolates (78%, 39/50), followed by *bla*_NDM_ (48%, 24/50), *bla*_IMP_ (22%, 11/50), *bla*_VIM_ (12%, 6/50), and *bla*_KPC_ (8%, 4/50) [31]. In the present work, among the 12 CRKP isolates we examined, 7 isolates harbored the *bla*_KPC_ gene and 1 isolate carried the *bla*_OXA-48_ gene. Our observations were in agreement with the previous work conducted in Taiwan, with also *bla*_KPC_ as the dominant gene [28].

Since carbapenem resistance worsened, novel therapeutic options were urgently requested to address the critical public health issue [35]. To this end, some β-lactam-β-lactamase inhibitor combination therapies were introduced into clinical settings recently, such as meropenem–vaborbactam, imipenem–relebactam, and ceftazidime–avibactam [7,8,9,10]. Moreover, combinations of azidothymidine (AZT), an antiviral agent usually used against HIV infection, plus other agents, were also reported, including AZT–colistin and AZT–fosfomycin [11,12,13]. Hu et al. obtained 74 antibiotic-resistant Enterobacteriaceae strains from hospitals in 9 countries, including 23 ESBL-producing *E. coli*, 31 ESBL, 7 NDM-1-producing strains, and 13 *mcr*-1-positive *E. coli* [11]. The checkerboard analysis revealed 60.9% (14/23), 87.1% (27/31), 100% (7/7), and 92.31% (12/13) synergetic effects in ESBL-producing E. coli, ESBL *K. pneumoniae*, NDM-1-producing strains, and *mcr*-1-positive E. coli, respectively. In another evaluation conducted in Taiwan, Chang et al. reported 100% synergistic activities for both KPC-producing and non-KPC-producing colistin-resistant CRKP isolates [12]. In vivo results demonstrated an extended lifespan and decreased risk ratios in a *Caenorhabditis elegans* model infected by a KPC-producing colistin-resistant CRKP isolate, positive support that AZT–colistin possessed the potential for treating CRKP infections. In a separate work, fosfomycin was noticed to have synergistic effects with AZT against 16 CRKP isolates, with the FICI of 87.5% isolates (14/16) ≤ 0.5 [13]. In vivo, the larvae of *Galleria mellonella* were infected by KPC-producing or NDM-producing CRKP isolate and treated with AZT–fosfomycin combination therapy. Compared to AZT or fosfomycin single-agent therapy, greater survival curves were observed for AZT–fosfomycin combination therapy, with 20% to 50% higher survival rates. In our work, significantly decreased MIC distributions were observed for AS101–AZT combination (both *p* < 0.0001), with 100% of synergistic activities against 12 CRKP isolates.

## 4. Materials and Methods

### 4.1. Bacteria Isolates

Twelve isolates of carbapenem-resistant *K. pneumoniae* (CRKP) were collected between 2013 and 2014 from 5 hospitals in a nationwide surveillance in Taiwan [28]. Carbapenem resistance was defined as being resistant to one of the carbapenems in accordance with the criteria recommended by the Clinical and Laboratory Standards Institute (CLSI) [36]. Among the 12 isolates, the main isolation source was urine (5/12, 41.7%), followed by sputum (4/12, 33.3%), abscess (1/12, 8.3%), blood (1/12, 8.3%), deep wound (1/12, 8.3%) (Table 4). The CRKP isolates were stored at –80 °C. Before the experiment, the isolates were recovered onto blood agar plate (Creative Media Plate, New Taipei, Taiwan) and cultured in the incubator at 37 °C. Colonies were subcultured onto fresh blood agar plate for stabilizing the physiological characteristics.

### 4.2. Antimicrobial Susceptibility

Susceptibilities of 19 antimicrobial agents against the 12 isolates were determined with a standard broth microdilution method following CLSI guidelines [36]. Briefly, two-fold serial dilutions of 19 antimicrobial agents were prepared. The bacterial suspension containing 2 × 10^8^ CFU/mL of cells was prepared in brain heart infusion (BHI) broth (Becton Dickinson (BD), Sunnyvale, CA, USA) and diluted for 200 folds. The well-prepared drug solution and the diluted bacterial suspension were added into 96-well plate, respectively. The final bacterial density was 5 × 10^5^ CFU/mL. After 16–18 h incubation, the result was detected by SpectraMax Absorbance Reader (CMax Plus, Molecular Devices, LCC, Sunnyvale, CA, US) at 600 nm. Among the 19 agents examined in this study, five classes of antibiotics were involved in the examination, including β-lactams and monobactam (ampicillin, ceftazidime, cefazolin, cefepime, cefoxitin, imipenem, ceftriaxone, meropenem, doripenem, ertapenem, cefotaxime, piperacillin–tazobactam, and aztreonam), aminoglycosides (amikacin and gentamicin), quinolones (ciprofloxacin and levofloxacin), inhibitors for folate synthesis (trimethoprim/sulfamethoxazole), and tetracycline (tigecycline). The results were also interpreted according to the breakpoints indicated by CLSI [36], and the interpretation for tigecycline was based on the guideline recommended by EUCAST [37]. Furthermore, the MICs for AS101 and azidothymidine (AZT) were also determined using the aforementioned broth microdilution method, and the results were collected for further evaluating the combination therapy of AS101 and AZT.

### 4.3. Synergistic Analysis

To investigate the synergistic effects between AZT (Toronto Research Chemicals, Toronto, Canada) and AS101 (Development Center for Biotechnology, Taipei, Taiwan), a checkerboard method was applied as described in a previous study [12,38]. In short, bacterial suspension containing 2 × 10^8^ CFU/mL of cells was prepared in brain heart infusion (BHI) broth (Becton Dickinson (BD), Sunnyvale, CA, USA) and subsequently diluted to an appropriate concentration. Two-fold serial dilution of AZT and AS101 were also prepared in the appropriate ranges of concentrations, and the well-prepared AZT and AS101 solutions (50 μL each) were added into a 96-well plate. One hundred μL of the diluted bacterial suspension was also dispensed into wells at a final bacterial concentration of 5 × 10^5^ CFU/mL. The result was also detected by SpectraMax Absorbance Reader (CMax Plus, Molecular Devices, LCC, Sunnyvale, CA, US) at 600 nm. The fractional inhibitory concentration (FIC) index was calculated to determine the synergistic effects as the following formula
$$\mathrm{FIC}=\frac{{\mathrm{MIC}}_{\mathrm{AS}101}\;\mathrm{in}\;\mathrm{combination}}{{\mathrm{MIC}}_{\mathrm{AS}101}\;\mathrm{in}\;\sin\mathrm{gle}}+\frac{{\mathrm{MIC}}_{\mathrm{ZDV}}\;\mathrm{in}\;\mathrm{combination}}{{\mathrm{MIC}}_{\mathrm{ZDV}}\;\mathrm{in}\;\sin\mathrm{gle}}$$

Generally, the interaction between two agents was defined as synergism if FIC index ≤ 0.5; it was interpreted as interaction while the FIC index was between 0.5 and 4; antagonism was defined as an FIC index > 4.

### 4.4. Polymerase Chain Reaction Detection

Extended-spectrum β-lactamase (ESBL) genes (*bla*_SHV_, *bla*_TEM_, *bla*_OXA_, *bla*_CTX-M-G1_, *bla*_CTX-M-G2_, and *bla*_CTX-M-G9_), plasmid-mediated AmpC genes (*bla*_DHA_ and *bla*_CMY_), carbapenemase genes (*bla*_KPC_, *bla*_NDM_, *bla*_IMP_, *bla*_NMC_, *bla*_SME_, *bla*_VIM_, *bla*_SPM-1_, *bla*_GIM-1_, *bla*_SIM-1_, *bla*_IMI_, *bla*_GES_, and *bla*_OXA-48_), plasmid-mediated colistin-resistant gene (*mcr*-1) and outer membrane porin genes (*omp*K35 and *omp*K36) were detected as in our previous study [12]. All the polymerase chain reactions were performed using TaKaRa Taq^TM^ (Cat. R001A, Takara Shuzo Co., Ltd., Tokyo, Japan) according to the instruction manual. Lastly, electrophoresis was applied to check PCR reactions, and DNA sequencing was utilized to validate the results, serviced by Genomics BioSci & Tech Co., Ltd. (New Taipei, Taiwan).

### 4.5. Statistical Analyses

The MIC distribution graphs of AS101 and AZT, in combination or alone, were constructed by GraphPad Prism (v9.0, CA, USA) and analyzed with paired Student’s *t*-test.

## 5. Conclusions

In this study, the noticeable changes in the MIC values after combining two drugs and fractional inhibitory concentration (FIC) index ≤ 0.5 in all strains indicated the synergistic activities between AS101 and azidothymidine against 12 clinical carbapenem-resistant *K. pneumoniae* (CRKP) isolates, harboring the carbapenemase genes or ESBL genes. Taken together, our efforts provided a new insight to develop a novel therapeutic option. Even though the sample size was small, our work still provided a proof of concept for the combination of AZT and AS101. Further in vivo studies and large-scale evaluations are needed for AS101–AZT combination therapy in future study.

## Figures and Tables

**Figure 1 pathogens-10-01552-f001:**
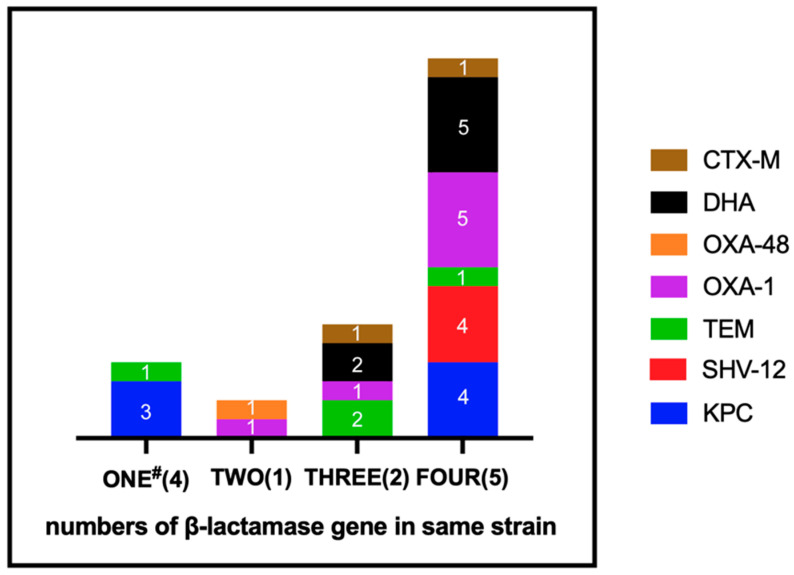
Distribution of the resistant mechanism among 12 isolates. #, number of resistant mechanisms on the same strain simultaneously (number of isolates).

**Figure 2 pathogens-10-01552-f002:**
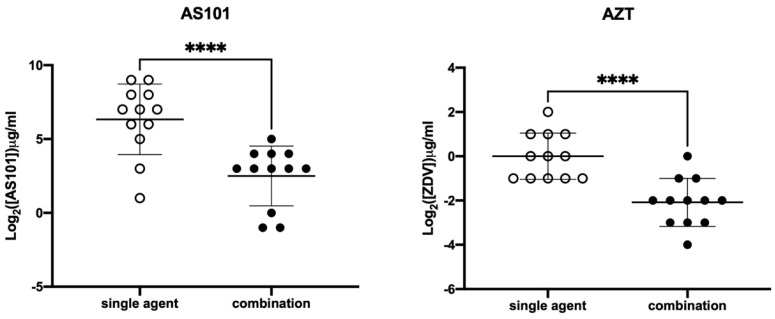
MIC distributions of AS101 (hollow circles), azidothymidine (AZT, hollow circles), and their combination (filled circles); bars in the middle of circles indicated means and standard deviations. ****, *p* < 0.0001 with the paired *t*-test.

**Table 1 pathogens-10-01552-t001:** Antimicrobial susceptibilities for 19 antimicrobial agents.

Antimicrobial Agent	Antibiotic Susceptibility ^1^		
	S	I	R
Amikacin	66.7%	0.0%	33.3%
Ampicillin	0.0%	0.0%	100.0%
Aztreonam	16.7%	0.0%	83.3%
Ceftazidime	0.0%	8.3%	91.7%
Cefazolin	0.0%	0.0%	100.0%
Ciprofloxacin	0.0%	0.0%	100.0%
Cefepime	0.0%	16.7%	83.3%
Cefoxitin	0.0%	0.0%	100.0%
Ceftriaxone	0.0%	0.0%	100.0%
Cefotaxime	8.3%	0.0%	91.7%
Gentamicin	41.7%	0.0%	58.3%
Imipenem	0.0%	0.0%	100.0%
Levofloxacin	0.0%	0.0%	100.0%
Meropenem	0.0%	0.0%	100.0%
Trimethoprim/Sulfamethoxazole	8.3%	0.0%	91.7%
Piperacillin–Tazobactam	8.3%	0.0%	91.7%
Ertapenem	0.0%	0.0%	100.0%
Tigecycline	91.7%	8.3%	0.0%
Doripenem	0.0%	0.0%	100.0%

^1^, abbreviations: S, susceptible; I, intermediate-resistant; R, resistant.

**Table 2 pathogens-10-01552-t002:** MIC values range, MIC_50_, MIC_75_, MIC_90_ for AS101, azidothymidine, and AS101–azidothymidine combination.

Agent	MIC Alone * (μg/mL)				MIC in Combination ^#^ (μg/mL)			
	Range	MIC_50_	MIC_75_	MIC_90_	Range	MIC_50_	MIC_75_	MIC_90_
AS101	2–512	128	256	512	0.5–32	8	16	16
AZT	0.5–4	1	2	2	0.0625–1	0.25	0.25	0.5

Abbreviations: MIC, minimum inhibitory concentration; AZT azidothymidine; MIC_50_, 50th percentile of MIC; MIC75, 75th percentile of MIC; MIC_90_, 90th percentile of MIC. *, MIC for AS101 or AZT in single agent; ^#^, MIC for combination of AS101 and AZT.

**Table 3 pathogens-10-01552-t003:** MIC values and FICI of 12 isolates against AS101, azidothymidine, AS101–azidothymidine combination.

Strains	MIC Alone * (μg/mL)		MIC in Combination ^#^ (μg/mL)		FICI	Interpretation
	AS101	AZT	AS101	AZT	(AS101, AZT)	
CRE-918	128	0.5	0.5	0.25	0.5	synergistic
CRE-949	256	1	8	0.25	0.28	synergistic
CRE-1017	64	2	8	0.5	0.38	synergistic
CRE-1038	256	0.5	16	0.125	0.31	synergistic
CRE-1044	512	0.5	16	0.0625	0.16	synergistic
CRE-1085	32	2	8	0.25	0.38	synergistic
CRE-1086	128	2	16	0.5	0.38	synergistic
CRE-1125	2	1	0.5	0.25	0.5	synergistic
CRE-1136	128	1	32	0.25	0.5	synergistic
CRE-1290	64	0.5	8	0.125	0.38	synergistic
CRE-1382	512	0.5	8	0.125	0.27	synergistic
CRE-1536	8	4	1	1	0.38	synergistic

Abbreviations: MIC, minimum inhibitory concentration; AZT, azidothymidine; FICI, fractional inhibitory concentration index. *, MIC for AS101 or AZT in single agent; ^#^, MIC for combination of AS101 and AZT.

**Table 4 pathogens-10-01552-t004:** Isolation sources of 12 isolates.

Isolation Source	No. of Isolates (%)
Abscess	1 (8.3)
Blood	1 (8.3)
Sputum	4 (33.3)
Urine	5 (41.7)
Deep wound	1 (8.3)

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
