# Peer review of "Synergistic Combination of AS101 and Azidothymidine against Clinical Isolates of Carbapenem-Resistant Klebsiella pneumoniae"

_pathogens, 2021, doi:10.3390/pathogens10121552_

Round 1
Reviewer 1 Report
The present manuscript investigate the use of two combined compounds, AS101 and azidothymidine against clinical isolates of carbapenem-resistant K. pneumoniae.
Even if the subject is extremely interesting due to the increasing concerns regarding the antibiotic resistance, which represent an health problem and highlight the urgent need of more studies on molecules used in combination with known antimicrobial molecules not only to avoid resistance but also for its worrying side effect,, there are some mayor issues that need to be clarified before it may be accepted for publication.
- Some weakness in writing needs to be amended; there are some typos
- The material is not presented in a proper way to allow easy comprehension.
- The introduction is not clear and need additional data to support the claims
- Please explain how the collected clinical isolates are maintained in laboratory and propagated for the experiments
- It could be appropriate to better explain MIC and combination of compounds (Tab 2 and Tab 3) not only in the captions but also in the text; details on the number of experiments performed are needed.
- Results have to elaborate on analysis of resistant mechanism, explaining also Figure 1 properly.
- Please explain fig 2, figure legends need more information
- Proper attention to statistical rigor is needed
- Despite the large amount of work, the authors should better discuss the collected results because in some points they generate confusion.
- Need in vivo validation to demonstrate functional relevance of these in vitro combination, or studies based on cytotoxicity in mammalian cells.
Author Response
Reviewer 1’s comments:
The present manuscript investigate the use of two combined compounds, AS101 and azidothymidine against clinical isolates of carbapenem-resistant K. pneumoniae.
Even if the subject is extremely interesting due to the increasing concerns regarding the antibiotic resistance, which represent a health problem and highlight the urgent need of more studies on molecules used in combination with known antimicrobial molecules not only to avoid resistance but also for its worrying side effect, there are some mayor issues that need to be clarified before it may be accepted for publication.
Reply: Thanks for the reviewer’s comment. The following is our response for each comment. We had corrected or adapted the part mentioned by reviewer 1. Please check and feel free to contact us if any question.
- Some weakness in writing needs to be amended; there are some typos
Reply: We are sorry for those typos. We revised all of the gene’s names and species names which should be italic but not italic before.
- The material is not presented in a proper way to allow easy comprehension.
Reply: We appreciate for the comment. The detailed information of materials was added in this manuscript in the section of material and method, such as “After 16~18 hours incubation, the result was detected by SpectraMax Absorbance Reader (CMax Plus, Molecular Devices, LCC, Sunnyvale, California, US) at 600 nm.” (335-337). “To investigate the synergistic effects between AZT (Toronto Research Chemicals, Toronto, Canada) and AS101 (Development Center for Biotechnology, Taipei, Taiwan)” (349-350), and “In short, bacterial suspension containing 2×108 CFU/ml of cells was prepared in brain heart infusion (BHI) broth (Becton Dickinson [BD], Sunnyvale, CA, USA) and subsequently diluted to an appropriate concentration.” (351-354)
- The introduction is not clear and need additional data to support the claims
Reply: We appreciated reviewer’s comments. We added a paragraph about the synergestic effect of AZT with several clinical antimicrobial agent, in line 58-62” In recent years, some novel β-lactam-β-lactamase inhibitor combination therapies were utilized in clinical, for instance, meropenem-vaborbactam, imipenem-relebactam, and ceftazidime-avibactam [7-10]. Besides, azidothymidine was also be reported its synergistic activities with some clinical antimicrobial agent, such as colistin and fosfomycin. [11-13].”
- Please explain how the collected clinical isolates are maintained in laboratory and propagated for the experiments
Reply: Thanks for this comment. Our procedure of maintaining the clinical isolates had been added in line 311-314: ” The CRKP isolates were stored at -80℃. Before the experiment, the isolates were recovered onto blood agar plate (Creative Media Plate, New Taipei, Taiwan) and culture in the incubator in 37℃. Colonies were subcultured onto fresh blood agar plate for stabilizing the physiological characteristics.”
- It could be appropriate to better explain MIC and combination of compounds (Tab 2 and Tab 3) not only in the captions but also in the text; details on the number of experiments performed are needed.
Reply: We are thankful for this comment. The interpretation of “MIC in alone” and “MIC in combination” had been added in line 201-203:” *, MIC for AS101 or AZT in single agent; #, MIC for combination of AS101 and AZT”. Besides, the numbers of experiment performed were added in line 198-199:” All of the checkerboard methods for checking synergistic effect were preformed three times repetition, and all results were reproducible.”
- Results have to elaborate on analysis of resistant mechanism, explaining also Figure 1 properly.
Reply: We are grateful for this comment. The subtitle of 2.2 was changed into “Detection of resistance gene”, because the original subtitle may be misunderstood. Also, we revised the interpretation of “2.2 Detection of resistance gene” in line 145-148: “Nevertheless, only 1 of the isolates (1/12, 8.3%) had two types of resistant gene (blaOXA-48 and blaOXA-1). 2 of isolates (2/12, 16.7%) harbored three types of resistant gene, including either blaOXA-1 or blaCTX-M in one of the isolates, and blaDHA and blaTEM in both.”
- Please explain fig 2, figure legends need more information
Reply: We appreciate for this comment. The explanation was added in line 213:” bars in the middle of circles indicated means and standard deviations.”
- Proper attention to statistical rigor is needed
Reply: Thanks for this comment. We had checked each statistical data in this manuscript. All the statistical methods were correct and suitable.
- Despite the large amount of work, the authors should better discuss the collected results because in some points they generate confusion.
Reply: We are grateful for the comment. We had adapted the paragraph discussing about Figure 1 in line 145-148” Nevertheless, only 1 of the isolates (1/12, 8.3%) had two types of resistant gene (blaOXA-48 and blaOXA-1). 2 of isolates (2/12, 16.7%) harbored three types of resistant gene, including either blaOXA-1 or blaCTX-M in one of the isolates, and blaDHA and blaTEM in both.” Besides, we also found a mistake in “2.2 Detection of resistant gene”. There were 5 isolates found with four types of resistant gene, not five types.
- Need in vivo validation to demonstrate functional relevance of these in vitro combination, or studies based on cytotoxicity in mammalian cells.
Reply: Thank you for the reviewer’s comment. AZT was a clinical agent for HIV infection, which was introduced in line 62-64:” Azidothymidine (AZT), a thymidine analog, possesses an antiretroviral activity and clinically treats patients with human immunodeficiency virus (HIV) infections, the acquired immunodeficiency syndrome (AIDS) [14].”. The 50% cytotoxicity of AS101 against Vero cell had also been reported as 145μg/mL, discussed in line 76-77” AS101 was used to treat autoimmune diseases, inflammatory bowel disease (IBD), multiple sclerosis (MS) and psoriasis [19], and some of the applications were ongoing clinical trials.” The previous efforts have proved that both AS101 and AZT are safe to apply in clinical settings. Our findings demonstrated an in vitro synergistic activity for the combination of AS101 and AZT, whereas the in vivo studies of combination AZT and AS101 must be necessary in the future study.” (line 281-282)
Reviewer 2 Report
Reviewer's report
Synergistic combination of AS101 and azidothymidine against 2 clinical isolates of carbapenem-resistant Klebsiella pneumoniae
Carbapenems are drugs of choice in the treatment of infections caused by K. pneumoniae strains ESBL+. However, the use or misuse of such antibiotics has contributed to the appearance of isolates resistant to carbapenems. Hence, searching for new solutions in patient therapy is very important. The authors tested the effectiveness of azidothymidine and immunomodulator AS101 to CRKP isolates and they demonstrated a synergistic effect for compounds.
Comments:
- The paper is interesting and well written, but I think 12 CRKP isolates is not enough for the credibility of the study.
How many repetitions were used in the experiments? Information should be included.
- •The growth rates of the examined strains in the absence and presence of the compounds should be shown. It can be interesting to synergistic effect.
- Discussion should be modified, first part (p.5) could be more focused.
- P. 7/L: 253 PCR reaction (my suggestion: amplicons); … and DNA sequencing was utilized to validate the results”. – sequencing by ….? Service…
- Italic is ommited for Klebsiella pneumoniae species (p. 3/L:98; p.6/L:205; p.7/L:262), for E. coli (p.6/L:187, 189; for Caenorhabditis elegans (p6/L:193), for Galleria mellonella (p.6/L: 197)
- Italic is ommited for genes (p.3/L: 100-107); p.5/L: 168-169175-178; p. 7/L: 245-250
- Format of references is not uniform, (the titles of ref. ) uppercase or lowercase
Author Response
Reviewer's report
Synergistic combination of AS101 and azidothymidine against 2 clinical isolates of carbapenem-resistant Klebsiella pneumoniae
Carbapenems are drugs of choice in the treatment of infections caused by K. pneumoniae strains ESBL+. However, the use or misuse of such antibiotics has contributed to the appearance of isolates resistant to carbapenems. Hence, searching for new solutions in patient therapy is very important. The authors tested the effectiveness of azidothymidine and immunomodulator AS101 to CRKP isolates and they demonstrated a synergistic effect for compounds.
Reply: We appreciated reviewer’s comments to help us to modified our article. The following is the response for each question from editor. Please check and feel free to contact us if any question.
Comments:
- The paper is interesting and well written, but I think 12 CRKP isolates is not enough for the credibility of the study.
Reply: Thanks for reviewer’s suggest. A large-scale evaluation is indeed necessary for synergistic effect of AZT and AS101. We added the description about the limitation in line 395-397: “Even though the sample size was small, our work still provided a proof of concept for the combination of AZT and AS101. Further in vivo studies and large-scale evaluations were needed for AS101-AZT combination therapy in future study.”
- How many repetitions were used in the experiments? Information should be included.
Reply: Many thanks to reviewer’s questions. All checkerboard methods for synergestic effect were preformed three times repetition. Interpretation was added in line 198-199:” All of the checkerboard methods for checking synergistic effect were preformed three times repetition, and all results were reproducible.”
- The growth rates of the examined strains in the absence and presence of the compounds should be shown. It can be interesting to synergistic effect.
Reply: Thanks for this comment. Due to its lowest FICI against the combination of AS101 and AZT, CRE-1044 was selected for the kinetic growth assay. The result was added as a supplementary material and described in line 207: “Additionally, synergestic effect was also observed in growth curve (Figure S1.)”.
- Discussion should be modified, first part (p.5) could be more focused.
Reply: We are grateful for the comment. we deleted a short paragraph to make the discussion more concise. Please check line 230-274: “Due to the transmission of resistant genes, the prevalence of carbapenem-resistant Enterobacteriaceae (CRE) has become a deadly threat to public health in the past years, causing severe infections associated with significant mortality [26]. Among approximately 140,000 cases of healthcare-associated Enterobacteriaceae infections in the United States yearly, around 9300 instances were caused by CRE, and 520 patients infected by CRE resulted in death [27]. The multidrug resistance was usually noticed among CRE isolates [28-31]. In a previous study conducted in the USA, a larger amount of CRKP isolates were collected from Jan 2014 to Mar 2015 in a long-term acute care hospital network [30]. The susceptibilities of selected antibiotics were examined against these isolates, including amikacin, ciprofloxacin, levofloxacin, gentamicin (or tobramycin), colistin (or polymyxin B), and tigecycline. The susceptibility to tigecycline was found to be highest (413/439, 94.1%), followed by colistin (or polymyxin B) (579/690, 83.9%), and amikacin (298/885, 33.7%). Low susceptibilities were revealed for ciprofloxacin (10/630, 1.6%), levofloxacin (12/713, 1.7%), and gentamicin (or tobramycin) (11/630, 1.7%). In Taiwan, Chiu et al. collected 457 isolates of CRKP from 21 hospitals between Jan 2012 and Aug 2015 [28], and the antimicrobial susceptibility testing of 19 agents against the 457 isolates was performed. Among agents they tested, only amikacin, colistin, and tigecycline showed high potencies, with susceptibilities of 78.8% (360/457), 85.6% (391/457), and 88.6% (405/457), respectively. According to a recent study from Iran, 50 isolates of CRKP were examined against 15 agents and were revealed with 85% susceptibility rate against only amikacin and high resistance rate against the other antimicrobial agents. [31]. In this study, tigecycline demonstrated the highest susceptibility (91.7%, 11/12) among 19 agents, followed by amikacin (66.7%, 8/12) and gentamicin (41.7%, 5/12), highlighting an urgent need for novel agents for CRKP.”
- 7/L: 253 PCR reaction (my suggestion: amplicons); … and DNA sequencing was utilized to validate the results”. – sequencing by ….? Service…
Reply: Thanks for this comment from the reviewer. The DNA sequencing was serviced by Genomics BioSci & Tech Co., Ltd (New Taipei, Taiwan). The related information was added in lines 372-373:” Lastly, electrophoresis was applied to check PCR reactions, and DNA sequencing was utilized to validate the results, serviced by Genomics BioSci & Tech Co., Ltd (New Taipei, Taiwan).”
- Italic is ommited for Klebsiella pneumoniaespecies (p. 3/L:98; p.6/L:205; p.7/L:262), for E. coli (p.6/L:187, 189; for Caenorhabditis elegans (p6/L:193), for Galleria mellonella (p.6/L: 197)
Reply: We appreciate this comment. We feel sorry for this mistake. All the species names had already been corrected. We also check the whole manuscript again.
- Italic is ommited for genes (p.3/L: 100-107); p.5/L: 168-169175-178; p. 7/L: 245-250
Reply: Thanks for the kind reminder. We apologized for the error sincerely. It may be some unexcepted changes in this file during submitting. All of the genes had already been turned into italic.
- Format of references is not uniform, (the titles of ref. ) uppercase or lowercase
Reply: We are sorry for the mistake. We had corrected the mistakes in the section reference.
Round 2
Reviewer 1 Report
Authors answered to all comments that I have committed about the manuscript.